# Mental Health Treatment Reported by US Workers before and during the COVID-19 Pandemic: United States (2019–2020)

**DOI:** 10.3390/ijerph20010651

**Published:** 2022-12-30

**Authors:** Ja K. Gu, Luenda E. Charles, Penelope Allison, John M. Violanti, Michael E. Andrew

**Affiliations:** 1Bioanalytics Branch, Health Effects Laboratory Division, National Institute for Occupational Safety and Health, Centers for Disease Control and Prevention, Morgantown, WV 26505, USA; 2Department of Epidemiology and Environmental Health, School of Public Health and Health Professions, University at Buffalo, State University of New York, Buffalo, NY 14214, USA

**Keywords:** prevalence, mental health treatment, workers, NHIS, COVID-19

## Abstract

The COVID-19 pandemic introduced a significant and unprecedented exacerbation of community mental health challenges. We compared the prevalence of mental health treatment (MHT) before and during the COVID-19 pandemic among US workers. Self-reported MHT data (N = 30,680) were obtained from the Sample Adult data of the National Health Interview Survey (2019 and 2020). MHT was defined as having taken prescription medications for mental health issues or receiving counseling from a mental health professional in the past 12 months. We calculated age-adjusted prevalence estimates and employed *t*-tests to compare MHT in 2019 and 2020 using SAS-callable SUDAAN 11.0. The prevalence of MHT significantly increased from 16.3% in 2019 to 17.6% in 2020 (difference = 1.3, *p* = 0.030). The prevalence of taking prescription medications for mental health issues significantly increased in 2020 compared to 2019 (12.5% to 13.6%, difference = 1.1, *p* = 0.037). The prevalence of receiving counseling significantly increased but only among those who worked 30–49 h/week, difference = 1.2, *p* = 0.022. US workers, especially those with typical work hours, appeared to experience higher mental distress during the first year of the pandemic compared to the year prior to the pandemic period. These findings highlight the need for targeted interventions to address mental health issues in these workers.

## 1. Introduction

Mental health disorder is a major public health burden which is made worse by that fact that affected persons usually need additional care for acute or chronic physical illnesses [1]. The economic burden for treating persons with these mental health disorders is huge and the cost may rise to billions of dollars per year [2]. The COVID-19 (coronavirus disease 2019) pandemic has been associated with mental health challenges including depression, anxiety, and other psychological disorders [3]. This may be due to fear of disease spread, social isolation, economic strain, and future uncertainty. Recent studies showed that symptoms of anxiety, depression, psychological distress, and mental health treatment (MHT) among US adults increased during the early months of 2020 compared to previous years [3,4]. In one of those studies, it was reported that the prevalence of symptoms of psychological distress among US adults was dramatically higher in 2020 (13.6%) than in 2018 (3.9%) [3]. This prevalence has continued to increase. For example, data from the Household Pulse Survey showed increases in anxiety or depressive disorder symptoms from August 2020 to February 2021 [5]. Results from other studies showed that the prevalence of any MHT among US adults increased during 2020 (20.3%) compared to 2019 (19.2%) [6,7]. Furthermore, the prevalence of taking prescription medications for mental health issues increased from 15.8% to 16.5% by 2020 and receiving professional counseling or therapy increased from 9.5% to 10.1% by 2020.

We were able to identify two studies that investigated the adverse outcomes associated with the COVID-19 pandemic in US workers. In the first study which was conducted among employed and unemployed US adults during June 2020, over 40.9% of respondents reported at least one mental health condition [8]. The results showed that the prevalence of symptoms of anxiety or depressive disorder was 42% higher among essential workers (health care personnel, first responders, and grocery store workers) compared to nonessential workers in 2020 [8]. Participants who reported mental health treatment for diagnosed anxiety or depression also reported higher prevalence of symptoms of adverse mental and behavioral health conditions compared with those who did not.

In the second study, investigators used data from the 2020 Health, Ethnicity, and Pandemic Study to study persons who were employed before the COVID-19 pandemic. Their results showed that, compared to workers with no change in employment during the pandemic, those who experienced permanent job loss had the highest prevalence of psychological distress. This finding was especially evident among non-Hispanic Black and Asian workers [9].

The findings of these two studies motivated us to contribute to the literature on the topic of COVID-19-related mental health outcomes among US workers. Therefore, the aims of this study were to compare the prevalence, over the previous 12 months, among US workers who had (1) taken prescription medications for mental health, (2) received counseling or therapy from a mental health professional, or (3) any MHT, which includes taking prescription medications or receiving counseling, before (2019) and during (2020) the COVID-19 pandemic.

## 2. Methods

### 2.1. Data Source: Sample Adult Data (2019–2020)

Data were obtained from the 2019 and 2020 National Health Interview Survey (NHIS), conducted by the CDC’s National Center for Health Statistics (NCHS). The NHIS is a nationally representative annual household survey of the civilian non-institutionalized US population. Interviews are usually conducted in person with follow-ups via telephone. However, due to the COVID-19 pandemic, data collection from April through June were conducted by telephone only, and from July to December 2020 by telephone first with follow-ups to complete interviews by personal visit. The response rates for the 2019 Sample Adult and 2020 Sample Adult Partial data were 59.1% and 48.9%, respectively. Extensive details about the questionnaire, methodology, data, and documentation are available at the NCHS website [10]. The total number of adult participants for the Sample Adult data in 2019–2020 was 53,150. From this number we included 30,680 workers in this study and excluded 22,470 persons who either did not work or did not respond to the question “Did you work in the last week.” NHIS is approved by the Research Ethics Review Board of the National Center for Health Statistics and the US Office of Management and Budget. All NHIS respondents provided oral consent prior to participation.

### 2.2. Mental Health Treatment

Participants were first asked “How often do you feel worried, nervous, or anxious?” If they said daily, weekly, monthly, or a few times a year, they were then asked “Do you take prescription medication for these feelings?” They were asked “How often do you feel depressed?” with a similar follow-up question about prescription medication. The other question was “During the past 12 months, did you take prescription medication to help you with any other emotions or with your concentration, behavior, or mental health?” Participants who responded ‘yes’ to any of these three questions were considered to have taken medication for their mental health.

Participants were considered to have received counseling or therapy if they responded ‘yes’ to the question, “During the past 12 months, did you receive counseling or therapy from a mental health professional such as a psychiatrist, psychologist, psychiatric nurse, or clinical social worker?”

Participants were considered to have received any MHT if they reported having taken medication for their mental health, received counseling or therapy from a mental health professional, or both in the past 12 months.

### 2.3. Sociodemographic and Work Characteristics

Sociodemographic and job characteristics were self-reported and included sex, age, race/ethnicity, education level, marital status, number of children in family, family income, living alone (yes/no), urbanization level, and work hours. Since the rate for ‘no response’ to the income question was high, family income was analyzed using the multiple imputation method to produce efficient estimates. Urbanization level was defined with metropolitan size and status which was determined using the 2013 NCHS urban-rural classification scheme for counties. Our categories were ‘large metropolitan’ which included large central and large fringe metropolitan counties, ‘medium or small metropolitan’ which included medium and small metropolitan counties, and ‘non-metropolitan’ which included micropolitan and non-core counties. We categorized working hours as follows: 1–29 h per week as shorter hours, 30–49 h per week as typical hours, and 50+ hours per week as longer hours.

### 2.4. Data Analysis

To generate a nationally representative estimate for the complex survey design, all prevalence estimates (%) were weighted using the weight variable that divides by 2 the 2019 Sample Adult weight and the 2020 Sample Adult partial weight, and they were age-adjusted using the standardized distribution of age among adult workers in NHIS 2019 and 2020. Standard errors were estimated using Taylor series linearization methods. SAS callable SUDAAN v11 was used for statistical analyses. All reported *p*-values were two-sided and a *p*-value of <0.05 was considered statistically significant.

## 3. Results

The sample sizes and weighted percentages for selected characteristics of the study sample in 2019 and 2020 are shown in Table A1 in Appendix A. The proportions of most of the characteristics were not significantly different between 2019 and 2020. However, some significant differences were observed for educational level, marital status, and family income.

The proportion of US workers who received any form of MHT increased by 1.3 from 2019 to 2020 (16.3% to 17.6%, *p* = 0.030) (Table 1). The prevalence of taking prescription medications for mental health issues significantly increased (12.5% to 13.6%, diff. = 1.1, *p* = 0.037), while that of receiving counseling or therapy from a mental health professional slightly increased (but not significantly) during the same period (8.6% to 9.3%, diff. = 0.7, *p* = 0.103).

Change in the percentage of any MHT significantly increased among workers aged 30–54 years (16.6% to 18.4%, diff. = 1.8, *p* = 0.016), but not among workers aged 18–29 years and 55+. The difference in the prevalence of receiving any MHT between 2019 and 2020 was significant among non-Hispanic White workers (20.8% to 22.3, diff. = 1.5, *p* = 0.041), while the differences among persons of other racial and ethnic groups (non-Hispanic Blacks, non-Hispanic Asians, and Hispanics) were not significantly different. Workers who were married reported significant increases from 2019 to 2020 both in taking prescription medications (11.9% to 13.4%, diff. = 1.5, *p* = 0.022) and receiving counseling (7.2% to 8.6%, diff. = 1.4, *p* = 0.009).

The prevalence of taking prescription medications increased significantly among workers with higher family incomes ($100K+) (11.6% to 13.4%, diff. = 1.8, *p* = 0.030). Among workers living in large metropolitan areas, the prevalence of obtaining any MHT increased (15.4% to 17.0%, diff. = 1.6, *p* = 0.033). Among those who worked the typical hours (30–49 h per week), the prevalence of receiving counseling increased between the two periods (7.8% to 9.0%, diff. = 1.2, *p* = 0.022). The prevalence of receiving counseling did not increase among workers with shorter hours (1–29 h per week, diff. = −0.2, *p* = 0.858) and workers with the longer hours (50+ hours per week, diff. = −0.4, *p* = 0.636).

## 4. Discussion

The COVID-19 pandemic has been associated with increased mental health burden among US workers. The prevalence of any MHT among US workers increased during 2020 compared to 2019, a finding consistent with previous studies in US adults [3,7]. Those who worked for 30–49 h per week during 2020 had a higher prevalence of receiving counseling for mental health compared to 2019. The absolute prevalence of MHT was higher in all adults compared to adult workers (19.2% vs. 16.3% in 2019; 20.3% vs. 17.6% in 2020). The protective factor of being employed (i.e., the Healthy Worker Effect) may have played a role in the lower prevalence of MHT among workers. For example, the difference in the prevalence of MHT between 2019 and 2020 was 1.3% among workers but 1.1% among adults [6,7]. Specifically, the change in the prevalence of taking prescription medications was 1.1% (12.5% to 13.6%) in workers and 0.7% (15.8% to 16.5%) in adults [6,7]. The change in the prevalence of receiving professional counseling in workers was similar to that for adults between 2019 and 2020 (9.5% to 10.1%, 0.6% in adults; 8.6% to 9.3%, 0.7% in workers).

Our results show that workers appeared to experience a greater psychological burden due to the COVID-19 pandemic. In the early weeks of the pandemic, before development of a vaccine, workers experienced major disruptions to their jobs. Not everyone was able to take advantage of telework, but they still needed to meet their financial obligations. As a result, workers were extremely stressed due to fears of becoming infected with COVID-19, frustrated with having to adhere to mandatory and useful public health policies of social distancing, masking, lockdowns, and quarantine, and concerns of job security and financial problems. These and other factors may explain the higher use of MHT among workers.

Consistent with previous studies [6,7], female workers had a higher prevalence of MHT than male workers. In contrast with previous studies, the prevalence of MHT did not change among two racial/ethnic groups, non-Hispanic Black and Hispanic workers. The prevalence of taking medication for mental health issues decreased among non-Hispanic Black workers.

While the prevalence of taking medications significantly increased, the prevalence of receiving professional counseling did not increase. Face-to-face counseling may not have increased due to fear of coronavirus infection from personal contact, although tele-counseling was expanding [11,12]. It is also possible that more workers may have preferred medication.

### 4.1. Limitations

Our findings are subject to several limitations. The difference in survey data collection methods may have resulted in response bias because response rates are typically higher for in-person interviews (2019) than for telephone interviews (2020), especially with sensitive topics [13]. Using the NHIS 2019–2020 Sample Adult datasets, we were not able to compare the prevalence within the socio-demographic subgroups. We would have needed to analyze the whole cross-sectional Sample Adults datasets, which was outside of our study goals. Due to limited occupational information in the 2019 NHIS survey, we could not compare the prevalence of MHT by occupational or industry group. Additionally, it is important to note that we cannot prove that the changes observed from 2019 to 2020 were caused by the COVID-19 pandemic. Without data from years before 2019 it is not possible to examine whether there was an increasing trend in mental health care utilization that started before 2019, and it is not possible to estimate the usual year to year variation in these measures. For this reason, it is not possible to rule out whether the observed changes from 2019 to 2020 arose from an existing trend or from random year to year variation.

### 4.2. Practical Implications

To improve the treatment of mental health during a pandemic, increased availability of and access to online counseling may be beneficial, especially if informed by the recent experiences of practitioners providing care during the pandemic [14]. Effective efforts to reduce stigma associated with mental health issues are also needed so more people may seek care. To solve issues around burnout, one author suggested that mental health awareness, a focus on increasing fairness, hybrid offerings and flexible hours, and personalizing communication are all initiatives that can be successfully implemented to address the mental health burden [15].

## 5. Conclusions

In a nationally representative sample of US working adults, the prevalence of taking prescription medications for mental health significantly increased across the study period (2019–2020), while the prevalence of receiving counseling or therapy did not change significantly. US workers, especially those who had the typical hours (i.e., 30–49 h), appeared to experience higher mental distress. These findings may inform efforts to address mental health needs in these workers during times of increased national or global stress.

## Figures and Tables

**Table 1 ijerph-20-00651-t001:** Age-adjusted prevalence of mental health treatment in pre-pandemic (2019) and pandemic (2020) years by characteristics among U.S. Workers.

	Mental Health Treatment (Medication + Counseling)				Medication for Mental Health				Counseling/Therapy for Mental Health			
Characteristics	2019 Prev. ± SE	2020 Prev. ± SE	Diff. ± SE	*p*-Value ^1^	2019 Prev. ± SE	2020 Prev. ± SE	Diff. ± SE	*p*-Value ^1^	2019 Prev. ± SE	2020 Prev. ± SE	Diff. ± SE	*p*-Value ^1^
All	16.3 ± 0.4	17.6 ± 0.5	1.3 ± 0.6	0.030	12.5 ± 0.3	13.6 ± 0.4	1.1 ± 0.5	0.037	8.6 ± 0.3	9.3 ± 0.4	0.7 ± 0.4	0.103
Age group (years)												
18–29	17.7 ± 0.8	19.3 ± 1.2	1.6 ± 1.3	0.228	12.1 ± 0.7	13.2 ± 1.0	1.1 ± 1.1	0.327	11.5 ± 0.6	12.4 ± 0.9	1.1 ± 1.1	0.328
30–54	16.6 ± 0.5	18.4 ± 0.6	1.8 ± 0.7	0.016	12.8 ± 0.4	14.4 ± 0.6	1.6 ± 0.6	0.022	8.9 ± 0.3	9.7 ± 0.4	0.8 ± 0.5	0.151
55+	14.4 ± 0.5	14.0 ± 0.8	−0.3 ± 0.9	0.708	12.2 ± 0.5	12.1 ± 0.7	−0.1 ± 0.2	0.850	5.1 ± 0.4	5.3 ± 0.4	0.2 ± 0.5	0.739
Gender												
Male	10.7 ± 0.4	11.9 ± 0.6	1.2 ± 0.7	0.089	7.9 ± 0.3	8.7 ± 0.5	0.8 ± 0.6	0.190	5.9 ± 0.3	6.7 ± 0.5	0.8 ± 0.6	0.156
Female	22.6 ± 0.6	23.9 ± 0.7	1.3 ± 0.9	0.151	17.6 ± 0.5	18.9 ± 1.0	1.3 ± 0.8	0.102	11.6 ± 0.4	12.2 ± 0.5	0.6 ± 0.7	0.344
Race/Ethnicity ^2^												
NH White	20.8 ± 0.5	22.3 ± 0.6	1.5 ± 0.8	0.041	16.2 ± 0.4	18.0 ± 0.6	1.8 ± 0.7	0.010	10.6 ± 0.3	11.0 ± 0.6	0.4 ± 0.6	0.448
NH Black	9.7 ± 0.8	10.0 ± 1.2	0.3 ± 1.4	0.806	7.2 ± 0.7	5.4 ± 0.8	−1.8 ± 1.0	0.062	6.1 ± 0.6	7.3 ± 1.0	1.2 ± 1.2	0.320
NH Asian	6.7 ± 0.9	7.6 ± 1.4	0.9 ± 1.7	0.605	4.5 ± 0.8	3.6 ± 0.9	−0.9 ± 1.1	0.452	4.9 ± 0.7	5.9 ± 1.3	1.0 ± 1.4	0.483
Hispanic	8.8 ± 0.6	8.9 ± 1.0	0.1 ± 1.2	0.964	6.3 ± 0.5	6.2 ± 0.8	−0.1 ± 0.9	0.909	4.7 ± 0.4	6.1 ± 0.8	1.4 ± 0.9	0.136
Education												
Grade: 0–12	12.3 ± 0.6	13.1 ± 0.8	0.8 ± 1.0	0.379	9.9 ± 0.5	10.5 ± 0.7	0.6 ± 0.8	0.441	5.2 ± 0.4	5.5 ± 0.5	0.3 ± 0.7	0.600
Some college	18.1 ± 0.6	18.8 ± 0.9	0.7 ± 1.0	0.507	14.6 ± 0.6	15.3 ± 0.8	0.7 ± 1.0	0.493	9.0 ± 0.5	9.2 ± 0.7	0.2 ± 0.8	0.772
Bachelor’s degree	18.8 ± 0.7	20.4 ± 0.9	1.6 ± 1.2	0.175	13.4 ± 0.6	15.2 ± 0.8	1.8 ± 1.0	0.063	11.3 ± 0.5	11.8 ± 0.7	0.5 ± 0.9	0.598
Master’s degree+	18.9 ± 0.9	21.2 ± 1.3	2.3 ± 1.6	0.152	12.9 ± 0.8	14.1 ± 1.1	1.2 ± 1.3	0.385	12.3 ± 0.8	14.6 ± 1.1	2.3 ± 1.3	0.086
Marital Status												
Married	15.1 ± 0.4	17.1 ± 0.6	2.0 ± 0.7	0.007	11.9 ± 0.4	13.4 ± 0.6	1.5 ± 0.7	0.022	7.2 ± 0.3	8.6 ± 0.5	1.4 ± 0.5	0.009
Widowed/Div.	23.9 ± 1.6	25.5 ± 2.6	1.6 ± 3.0	0.597	16.1 ± 1.2	20.9 ± 2.5	4.8 ± 2.7	0.078	16.0 ± 1.6	13.6 ± 2.3	−2.4 ± 2.9	0.407
Never married	16.9 ± 0.7	18.7 ± 1.0	1.8 ± 1.2	0.143	13.0 ± 0.7	14.0 ± 0.9	1.0 ± 1.1	0.368	10.4 ± 0.6	11.2 ± 0.8	0.8 ± 1.0	0.387
Living Alone												
Yes	20.0 ± 0.7	21.3 ± 1.0	1.3 ± 1.2	0.276	14.2 ± 0.6	15.8 ± 0.9	1.6 ± 1.1	0.126	12.7 ± 0.6	13.4 ± 0.9	0.7 ± 1.0	0.523
Live with other	15.6 ± 0.4	16.9 ± 0.5	1.3 ± 0.6	0.046	12.1 ± 0.4	13.1 ± 0.5	1.0 ± 0.6	0.078	7.8 ± 0.3	8.5 ± 0.4	0.7 ± 0.5	0.118
Family Income ^3^												
<35 K	16.8 ± 0.8	16.6 ± 1.2	−0.2 ± 1.3	0.890	13.3 ± 0.7	13.1 ± 1.0	−0.2 ± 1.2	0.866	8.7 ± 0.5	8.6 ± 0.9	−0.1 ± 1.0	0.903
35–<75 K	16.1 ± 0.6	16.8 ± 0.9	0.7 ± 1.0	0.475	12.9 ± 0.5	13.4 ± 0.8	0.5 ± 0.9	0.579	7.7 ± 0.4	8.5 ± 0.7	0.8 ± 0.8	0.319
75–<100 K	17.3 ± 0.9	17.7 ± 1.2	0.4 ± 1.5	0.765	13.1 ± 0.8	14.3 ± 1.1	1.2 ± 1.3	0.350	9.2 ± 0.7	8.8 ± 0.9	−0.4 ± 1.1	0.698
100 K+	16.0 ± 0.6	18.3 ± 0.9	2.3 ± 1.0	0.027	11.6 ± 0.5	13.4 ± 0.7	1.8 ± 0.8	0.030	9.0 ± 0.4	10.2 ± 0.6	1.2 ± 0.8	0.147
Urbanization level												
Large metro.	15.4 ± 0.5	17.0 ± 0.7	1.6 ± 0.8	0.033	11.4 ± 0.4	12.3 ± 0.5	0.9 ± 0.6	0.144	8.9 ± 0.3	10.4 ± 0.5	1.5 ± 0.6	0.015
Medium/small	17.8 ± 0.7	18.4 ± 0.9	0.6 ± 1.0	0.519	13.9 ± 0.6	14.8 ± 0.8	0.9 ± 0.9	0.288	8.9 ± 0.5	8.7 ± 0.6	−0.2 ± 0.7	0.748
Nonmetropolitan	17.2 ± 1.0	18.1 ± 1.6	0.9 ± 2.0	0.644	14.5 ± 0.9	16.3 ± 1.4	1.8 ± 1.8	0.294	6.5 ± 0.6	5.8 ± 0.9	−0.7 ± 1.2	0.528
Work hours (per week)												
1–29 h	21.8 ± 1.0	22.6 ± 1.4	0.8 ± 1.6	0.628	16.5 ± 0.9	17.5 ± 1.2	1.0 ± 1.4	0.484	12.4 ± 0.8	12.2 ± 1.0	−0.2 ± 1.2	0.858
30–49 h	15.4 ± 0.4	16.9 ± 0.6	1.5 ± 0.7	0.041	12.1 ± 0.4	13.0 ± 0.5	0.9 ± 0.6	0.150	7.8 ± 0.3	9.0 ± 0.4	1.2 ± 0.5	0.022
50+ h	15.1 ± 0.7	15.9 ± 1.0	0.8 ± 1.2	0.516	10.8 ± 0.6	12.1 ± 0.9	1.3 ± 1.1	0.241	8.5 ± 0.5	8.1 ± 0.7	−0.4 ± 0.8	0.636

^1^*p*-values were obtained using the T-test. ^2^ NH = Non-Hispanic. ^3^ Family Income was analyzed with multiple imputed datasets.

## Data Availability

The data presented in this study are available on request to qualified researchers from the corresponding author.

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
