# Peer review of "Mental Health Treatment Reported by US Workers before and during the COVID-19 Pandemic: United States (2019–2020)"

_ijerph, 2022, doi:10.3390/ijerph20010651_

Round 1

Reviewer 1 Report

The short communication “Mental health treatment reported by US workers before and 2 during the COVID-19 pandemic: United States (2019–2020)” is an interesting report.

In the Results section, I would suggest a minor detail to consider in the Appendix Table 1: in addition to presenting the Wg (%) the authors may consider adding one column for 2019 and one for 2020 showing the percentage of actual participants, which may be clearer for readers as to how the data compares to the larger data set (e.g., age group 18-29, with 3,394 individuals in 2019 would represent 18.0% of the total 18,808 individuals, and it would be easy to compare to the 24.1% Wg(%).

In Table 1, if I understand this correctly, there is mistake in line Work hours per week 30-49 where it is stated that there was a 1.5 increase, but it says it was 15.0 in 2019 and 16.0 in 2020, which would be a 1.0 increase. This needs to be fixed in the table.

In the Discussion section, the authors say: “While the absolute prevalence of MHT was higher in all adults compared to adult workers (19.2% vs. 16.3% in 2019; 20.3% vs. 17.6% in 2020), the mental health of workers appeared to be affected to a greater extent.” The increase in all adults is 1.1% and in adult workers is 1.3%, which does not look like a difference allowing authors to say that the mental health of workers appeared to be affected to a greater extent. It is also interesting that adult workers still had a lower prevalence of MHT compared to all adults. The protective factor of being employed (or other considerations affecting MHT prevalence among adult workers compared to all adults) would be interesting to mention.

Limitations, Suggestions, and Conclusion sections are fine.

Author Response

The short communication “Mental health treatment reported by US workers before and 2 during the COVID-19 pandemic: United States (2019–2020)” is an interesting report.

Authors’ response: We thank the reviewer for the many useful suggestions and comments.

 In the Results section, I would suggest a minor detail to consider in the Appendix Table 1: in addition to presenting the Wg (%) the authors may consider adding one column for 2019 and one for 2020 showing the percentage of actual participants, which may be clearer for readers as to how the data compares to the larger data set (e.g., age group 18-29, with 3,394 individuals in 2019 would represent 18.0% of the total 18,808 individuals, and it would be easy to compare to the 24.1% Wg(%).

 Authors’ response: Thank you for your suggestion. The NHIS is a complex survey design with oversampling of racial/ethnic minority subgroups (e.g., non-Hispanic blacks, Hispanics, and non-Hispanic Asians), low-income persons, and older adults. We feel that inclusion of the unweighted percentages into a table that’s already very busy would not add much clarity for readers. Instead, we decided to include a footnote to the table that provides the formula to calculate unweighted percentages.

In Table 1, if I understand this correctly, there is mistake in line Work hours per week 30-49 where it is stated that there was a 1.5 increase, but it says it was 15.0 in 2019 and 16.0 in 2020, which would be a 1.0 increase. This needs to be fixed in the table.

 Authors’ response: Thank you for catching this error. We have now corrected it. 

In the Discussion section, the authors say: “While the absolute prevalence of MHT was higher in all adults compared to adult workers (19.2% vs. 16.3% in 2019; 20.3% vs. 17.6% in 2020), the mental health of workers appeared to be affected to a greater extent.” The increase in all adults is 1.1% and in adult workers is 1.3%, which does not look like a difference allowing authors to say that the mental health of workers appeared to be affected to a greater extent. It is also interesting that adult workers still had a lower prevalence of MHT compared to all adults. The protective factor of being employed (or other considerations affecting MHT prevalence among adult workers compared to all adults) would be interesting to mention.

Authors’ response: We agree with the reviewer’s comment and have revised the statement “The absolute prevalence of MHT was higher in all adults compared to adult workers (19.2% vs. 16.3% in 2019; 20.3% vs. 17.6% in 2020). The protective factor of being employed (i.e., the Healthy Worker Effect) may have played a role in the lower prevalence of MHT among workers.     

Limitations, Suggestions, and Conclusion sections are fine.

 Authors’ response: Thank you!

Reviewer 2 Report

This is an interesting study, We will provide the authors with some suggestions for improving the manuscript. The comments will follow the structure and order of the manuscript.

The introductory section is too brief, it does not define the concept of mental health to which it refers. It should provide more studies and references on mental health and its scope during the pandemic.

Please remove authors' names when citing in the paper. Apply the citation rules of the journal. For example, look at line 43 …workers, and some were disproportionately affected. Czeisler and colleagues [6] found…

I have a doubt that the researchers say that 53,150 were surveyed and that only 30,680 were considered. It is important that the researchers clarify the reason or criteria for eliminating the rest of the sample. Please clarify the type of sampling that was done and the criteria by which these participants were chosen.

Section 2.2 Outcomes: Mental health treatment… is not adequate as the authors present it. They are not really resulted as the authors indicate. They simply state the description of the questions related to the selection criteria for considering study participants. This could be considered procedural in describing the sample selection. They are not the results as they are presented.

A specific instrument section is missing. The survey given to the participants would have psychometric properties (reliability, validity...) Adequately describe the instruments and their psychometric properties.

Do not describe the hypothesis contrasting tests that were used. They do not describe whether parametric tests or non-parametric tests were used.

The authors acknowledge that changes cannot be explained from the results because no pre-pandemic information is available. This is an important limitation. Without another outcome such as interviews ... it is difficult to sustain some of the authors' assertions and conclusions.

The introduction is very limited and does not focus on the topics of the study. They should provide data on the mental health of workers before the pandemic, they should delve into the relationship between mental health and social isolation. There are many studies on the effects of mental health during the pandemic. The literature references need to be increased and the study needs to be more adequately justified.

English needs to be edited by a native speaker.

Author Response

This is an interesting study, We will provide the authors with some suggestions for improving the manuscript. The comments will follow the structure and order of the manuscript.

Authors’ response: We thank the reviewer for the many useful suggestions and comments.

The introductory section is too brief, it does not define the concept of mental health to which it refers. It should provide more studies and references on mental health and its scope during the pandemic.

Authors’ response:  The Introduction was intentionally written to be brief because this manuscript was submitted as a ‘Short Communication’, not as an original research manuscript. Hence the reason we only briefly explained the current situation of psychological disorders and mental health treatment during the COVID-19 pandemic in the Introduction.

Please remove authors' names when citing in the paper. Apply the citation rules of the journal. For example, look at line 43 …workers, and some were disproportionately affected. Czeisler and colleagues [6] found…

Authors’ response:  We have now revised all citations and references in the paper.

I have a doubt that the researchers say that 53,150 were surveyed and that only 30,680 were considered. It is important that the researchers clarify the reason or criteria for eliminating the rest of the sample. Please clarify the type of sampling that was done and the criteria by which these participants were chosen.

Authors’ response: We revised the sentence “The total number of adult participants for the Sample Adult data in 2019-2020 was 53,150. From this number we included 30,680 workers in this study, and excluded 22,470 persons who either did not work or did not respond to the question ”Did you work in the last week.”

Section 2.2 Outcomes: Mental health treatment… is not adequate as the authors present it. They are not really resulted as the authors indicate. They simply state the description of the questions related to the selection criteria for considering study participants. This could be considered procedural in describing the sample selection. They are not the results as they are presented.

Authors’ response: Section 2.2 Outcomes: Mental health treatment’ under the methods section describes how the variable ‘Mental Health Treatment’ was assessed. These paragraphs are not part of our results.  Mental health treatment was also defined by the paper published by CDC (Terlizzi, 2020). In addition, our manuscript has been reviewed and approved by the National Center for Health and Statistics (NCHS), which is a CDC agency that collects and conducts the National Health Interview Survey (NHIS). We have decided to remove the word ‘Outcomes’ in the sub-title to prevent any further misinterpretation.

 Terlizzi EP, Zablotsky B. (2020). Mental health treatment among adults: United States, 2019. NCHS Data Brief, no 380. Hyattsville, MD: National Center for Health Statistics.

A specific instrument section is missing. The survey given to the participants would have psychometric properties (reliability, validity...) Adequately describe the instruments and their psychometric properties.

Authors’ response: The questions that are used in this study have already been tested and approved by the National Center for Health Statistics (NCHS) of the U.S. Centers for Disease Control and Prevention for use by researchers in the U.S. and throughout the world. We do not have the necessary information to describe the reliability and validity of these questions.

Do not describe the hypothesis contrasting tests that were used. They do not describe whether parametric tests or non-parametric tests were used.

Authors’ response: In this study, the authors only compared the prevalence of mental health treatment between 2019 and 2020. This is not an etiologic study.

The authors acknowledge that changes cannot be explained from the results because no pre-pandemic information is available. This is an important limitation. Without another outcome such as interviews ... it is difficult to sustain some of the authors' assertions and conclusions.

Authors’ response: We agree with the reviewer that is a limitation of this study and we have addressed this limitation in the Discussion (see below). However, we believe that these results are significant and still meaningful for public health in the United States.

Limitation: “Also, it is important to note that we cannot prove that the changes observed from 2019 to 2020 were caused by the COVID-19 pandemic. Without data from years before 2019 it is not possible to examine whether there was an increasing trend in mental health care utilization that started before 2019, and it is not possible to estimate the usual year to year variation in these measures. For this reason, it is not possible to rule out whether the observed changes from 2019 to 2020 arose from an existing trend or from random year to year variation”

The introduction is very limited and does not focus on the topics of the study. They should provide data on the mental health of workers before the pandemic, they should delve into the relationship between mental health and social isolation. There are many studies on the effects of mental health during the pandemic. The literature references need to be increased and the study needs to be more adequately justified.

Authors’ response: Thank you for your comments.  We agree that the Introduction is very limited. The reason is that this manuscript was submitted as ‘Short Communication’ and not as an original research manuscript.

English needs to be edited by a native speaker.

Authors’ response:  Thank you. Four of the five coauthors, who reviewed and revised the manuscript, are native English Language speakers. We have all again reviewed the entire manuscript. Can the reviewer point to the specific areas where additional editing is needed?

Reviewer 3 Report

Dear Authors,

congratulations on your manuscript which provides relevant data to understand the psychological burden of COVID pandemic on workers. It provides numbers which are really important to base any research. Here are just few suggestions. 

-in Discussion I suggest to provide some explanation or considerations about the fact that workers are more affected by pandemic in terms of psychological burden

-par. 4.2: I believe that "practical implications" suits better and I advice you to expand it a little more.

Good luck with your work.

Author Response

congratulations on your manuscript which provides relevant data to understand the psychological burden of COVID pandemic on workers. It provides numbers which are really important to base any research. Here are just few suggestions. 

Authors’ response: We thank the reviewer for the useful suggestions.

-in Discussion I suggest to provide some explanation or considerations about the fact that workers are more affected by pandemic in terms of psychological burden

Authors’ response: As suggested by the reviewer, we have added the following sentences.  “Our results show that workers appeared to experience a greater psychological burden due to the COVID-19 pandemic. In the early weeks of the pandemic, before development of a vaccine, workers experienced major disruptions to their jobs. Not everyone was able to take advantage of telework, yet they still needed to meet their financial obligations. As a result, workers were extremely stressed due to fears of becoming infected with COVID-19, frustrated with having to adhere to mandatory and useful public health policies of social distancing, masking, lockdowns, and quarantine, and concerns of job security and financial problems. These and other factors may explain the higher use of MHT among workers.”

-par. 4.2: I believe that "practical implications" suits better and I advice you to expand it a little more.

Authors’ response: Thank you.  We changed the sub-title to “4.2 Practical Implications”. In addition, we have added a sentence to this paragraph: “To solve issues around burnout, one author suggested that mental health awareness, a focus on increasing fairness, hybrid offerings and flexible hours, and personalizing communication are all initiatives that can be successfully implemented to address the mental health burden [12].”

Round 2

Reviewer 2 Report

I thank the authors for the changes they have made to the manuscript. However, there are still some aspects that need to be embroidered:

I thank the authors for the changes they have made to the manuscript. However, there are still some aspects that need to be embroidered:

For example, on line 33 it still appears: … McGinty and colleagues [1] reported…

Another example:

In line 43, the authors make the same mistake when they write: ...Czeisler and colleagues [6] found…

Please proofread the entire manuscript again and check that the citations follow the journal's guidelines.

The authors state that: There are few studies that focused on the mental health of US workers (vs. all adults) during the COVID-19 pandemic and also assessed MHT for 12 months so we sought to fill this research gap… It is necessary that they report what these few studies have provided in order to adequately justify the need for the present study.

The argument that this is a brief research report is not an argument that the introduction cannot be expanded. I am aware of the nature of your study.

Again, I recommend that the authors provide a brief extension to the introduction clarifying the concept of mental health to which they refer and citing more authors and studies on mental health in U.S. workers.

Another aspect to consider is the question of the link that the authors put in the method section.

Specifically, on line 68, the authors write: …documentation are available at https://www.cdc.gov/nchs/nhis/data-questionnaires documentation.htm

Please remove the link from the manuscript and cite it and include it in the bibliographic references. The text indicates only that the documentation is available from the National Center for Health Statistics.

When citing the National Center for Health Statistics in the bibliographic references include the link

Regards

Author Response

The responses to the reviewer's comments are listed below.  The attached document contains the revised manuscript with changes shown using the "Tracked changes" feature.

Thank you for your patience. Mr. Gu was on vacation and was unable to respond to these comments by the required deadline.

RESPONSES TO COMMENTS

Reviewer: I thank the authors for the changes they have made to the manuscript. However, there are still some aspects that need to be embroidered:

For example, on line 33 it still appears: … McGinty and colleagues [1] reported…

Another example:

In line 43, the authors make the same mistake when they write: ...Czeisler and colleagues [6] found…

Please proofread the entire manuscript again and check that the citations follow the journal's guidelines.

Authors’ response: Thank you again for your comments. We have removed the authors’ names. The revised sentences are shown using the “Track Changes” feature.

Reviewer: The authors state that: There are few studies that focused on the mental health of US workers (vs. all adults) during the COVID-19 pandemic and also assessed MHT for 12 months so we sought to fill this research gap… It is necessary that they report what these few studies have provided in order to adequately justify the need for the present study.

The argument that this is a brief research report is not an argument that the introduction cannot be expanded. I am aware of the nature of your study.

Again, I recommend that the authors provide a brief extension to the introduction clarifying the concept of mental health to which they refer and citing more authors and studies on mental health in U.S. workers.

Authors’ response: We were not able to cite more studies on mental health in U.S. workers because only found 2 published studies were found on the topic. As recommended, we have revised the Introduction and the revised sentences are shown using the “Track Changes” feature. We hope that we have adequately addressed the reviewer’s concerns.

Reviewer: Another aspect to consider is the question of the link that the authors put in the method section.

Specifically, on line 68, the authors write: …documentation are available at https://www.cdc.gov/nchs/nhis/data-questionnaires documentation.htm

Please remove the link from the manuscript and cite it and include it in the bibliographic references. The text indicates only that the documentation is available from the National Center for Health Statistics.

When citing the National Center for Health Statistics in the bibliographic references include the link

Authors’ response: Thank you for your suggestion. We have removed the link from the manuscript and included it in the Reference list.

Page 2: “Extensive details about the questionnaire, methodology, data, and documentation are available at the NCHS website [10].”
